# ANALYTICALLY TRACTABLE BAYESIAN DEEP Q-LEARNING

## ABSTRACT

Reinforcement learning (RL) has gained increasing interest since the demonstration it was able to reach human performance on video game benchmarks using *deep Q-learning* (DQN). The current consensus of DQN for training neural networks (NNs) on such complex environments is to rely on gradient-descent optimization (GD). This consensus ignores the uncertainty of the NN's parameters which is a key aspect for the selection of an optimal action given a state. Although alternative Bayesian deep learning methods exist, most of them still rely on GD and numerical approximations, and they typically do not scale on complex benchmarks such as the Atari game environment. In this paper, we present how we can adapt the temporal difference Q-learning framework to make it compatible with the *tractable approximate Gaussian inference* (TAGI) which allows estimating the posterior distribution of NN's parameters using a closed-form analytical method. Throughout the experiments with on- and off-policy reinforcement learning approaches, we demonstrate that TAGI can reach a performance comparable to backpropagation-trained networks while using only half the number of hyperparameters, and without relying on GD or numerical approximations.

## 1 INTRODUCTION

Reinforcement learning (RL) has gained increasing interest since the demonstration it was able to reach human performance on video game benchmarks using *deep Q-learning* (DQN) (Mnih et al., 2015; Van Hasselt et al., 2016). Deep RL methods typically require an explicit definition of an exploration-exploitation function in order to compromise between using the current policy and exploring the potential of new actions. Such an issue can be mitigated by opting for a Bayesian approach where we estimate the uncertainty of the neural network's parameters using Bayes' theorem, and select the optimal action given a state according to the parameter uncertainty using Thompson sampling technique (Strens, 2000). Bayesian deep learning (BDL) methods based on variational inference (Kingma et al., 2015; Hernández-Lobato & Adams, 2015; Blundell et al., 2015; Louizos & Welling, 2016; Osawa et al., 2019; Wu et al., 2019), Monte-Carlo dropout (Gal & Ghahramani, 2016), or Hamiltonian Monte-Carlo sampling (Neal, 1995) have shown to perform well on regression and classification benchmarks, despite being generally computationally more demanding than their deterministic counterparts. Note that none of these approaches allow performing the analytical inference for the weights and biases defining the neural network. Goulet et al. (2021) recently proposed the *tractable approximate Gaussian inference* (TAGI) method that allows estimating the posterior of the neural network's parameters using a closed-form analytical method. More specifically, TAGI leverages Gaussian conditional equations to analytically infer this posterior without the need of numerical approximations (i.e., sampling techniques or Monte Carlo integration) or optimization on which existing BDL methods rely. In addition, TAGI is able to maintain the same computational complexity as deterministic neural networks based on the gradient backpropagation. For convolutional architectures applied on classification benchmarks, this approach was shown to exceed the performance of other Bayesian and deterministic approaches based on gradient backpropagation, and to do so while requiring a smaller number of training epochs (Nguyen & Goulet, 2021).

In this paper, we propose a new perspective on probabilistic inference for Bayesian deep reinforcement learning which, up to now has been relying on gradient-based methods. More specifically, we present how can we adapt the temporal difference Q-learning framework (Sutton, 1988; Watkins & Dayan,

1992) to make it compatible with TAGI. Section 2 first reviews the theory behind TAGI and the expected value formulation through the Bellman's Equation. Then, we present how the action-value function can be learned using TAGI. Section 3 presents the related work associated with Bayesian reinforcement learning, and Section 4 compares the performance of the TAGI-DQN with the deterministic DQN on on- and off-policy RL approaches.

## 2 TAGI-DQN FORMULATION

This section presents how to adapt the DQN frameworks in order to make them compatible with analytical inference. First, Section 2.1 reviews the fundamental theory behind TAGI, and Section 2.1 reviews the concept of long-term expected value through the Bellman's equation (Sutton & Barto, 2018). Then, Section 2.3 presents how to make the Q-learning formulation (Watkins & Dayan, 1992) compatible with TAGI.

### 2.1 TRACTABLE APPROXIMATE GAUSSIAN INFERENCE

Bayesian deep learning aims to estimate the posterior of neural network's parameters $\boldsymbol{\theta}$ conditional on a given training dataset $\mathcal{D} = \{(\boldsymbol{x}_i, \boldsymbol{y}_i), \forall i \in 1 : \mathtt{D}\}$ using Bayes' theorem so that

$$f(\boldsymbol{\theta}|\mathcal{D}) = \frac{f(\boldsymbol{\theta})f(\mathcal{D}|\boldsymbol{\theta})}{f(\mathcal{D})}, \tag{1}$$

where $f(\boldsymbol{\theta})$ is the prior *Probability Density Function* (PDF) of parameters, $f(\mathcal{D}|\boldsymbol{\theta})$ is the likelihood and $f(\mathcal{D})$ is the marginal likelihood or evidence. Until recently, the posterior $f(\boldsymbol{\theta}|\mathcal{D})$ in Equation 1 has been considered to be intractable (Goodfellow et al., 2016; Goan & Fookes, 2020). TAGI has addressed this intractability issue by leveraging the Gaussian conditional equations; For a random vector of parameter $\boldsymbol{\theta}$ and observations $\boldsymbol{Y}$ for which the joint PDF satisfies

$$f\left(\begin{array}{c} \boldsymbol{\theta} \\ \boldsymbol{y} \end{array}\right) = \mathcal{N}\left(\left[\begin{array}{c} \boldsymbol{\theta} \\ \boldsymbol{y} \end{array}\right]; \left[\begin{array}{c} \boldsymbol{\mu_\theta} \\ \boldsymbol{\mu_Y} \end{array}\right], \left[\begin{array}{cc} \boldsymbol{\Sigma_\theta} & \boldsymbol{\Sigma_{Y\theta}^\intercal} \\ \boldsymbol{\Sigma_{Y\theta}} & \boldsymbol{\Sigma_Y} \end{array}\right]\right), \tag{2}$$

the Gaussian conditional equation describing the PDF of $\boldsymbol{\theta}$ conditional on $\boldsymbol{y}$ is defined following

$$\begin{aligned} f(\boldsymbol{\theta}|\boldsymbol{y}) &= \mathcal{N}(\boldsymbol{\theta}; \boldsymbol{\mu_{\theta|y}}, \boldsymbol{\Sigma_{\theta|y}}) \\ \boldsymbol{\mu_{\theta|y}} &= \boldsymbol{\mu_\theta} + \boldsymbol{\Sigma_{Y\theta}^\intercal} \boldsymbol{\Sigma_Y^{-1}}(\boldsymbol{y} - \boldsymbol{\mu_Y}) \\ \boldsymbol{\Sigma_{\theta|y}} &= \boldsymbol{\Sigma_\theta} - \boldsymbol{\Sigma_{Y\theta}^\intercal} \boldsymbol{\Sigma_Y^{-1}} \boldsymbol{\Sigma_{Y\theta}}. \end{aligned} \tag{3}$$

In its simple form, the intractable computational requirements for directly applying Equation 3 limits it to trivial neural networks. In order to overcome this challenge, TAGI leverages the inherent conditional independence structure of hidden layers $\boldsymbol{Z}^{(j-1)} \perp\!\!\!\perp \boldsymbol{Z}^{(j+1)} | \boldsymbol{z}^{(j)}$ under the assumptions that parameters $\boldsymbol{\theta}$ are independent between layers. This conditional independence structure allows breaking down the computations for Equation 3 into a layer-wise two-step procedure; *forward uncertainty propagation* and *backward update*.

The first forward uncertainty propagation step is intended to build the joint prior between the hidden states of a layer $j$, $\boldsymbol{Z}^{(j)}$, and the parameters $\boldsymbol{\theta}^{(j)}$ directly connecting into it. This operation is made by propagating the uncertainty from the parameters and the input layer through the neural network. In order to maintain the analytical tractability of the forward step, we must address two challenges: The first challenge is related to the product of the weights $W$ and activation units $A$ involved in the calculation of the hidden states $Z$ so that

$$Z_i^{(j+1)} = \sum_{k=1}^{\mathtt{A}} W_{i,k}^{(j)} A_k^{(j)} + B_i^{(j)}, \tag{4}$$

where $B$ is the bias parameter, $(j)$ refers to the $j^{th}$ layer, $i$ refers to a node in the current layer, $k$ refers to a node from the previous layer, and $\mathtt{A}$ is the number of hidden units from the previous layer. The second challenge is to propagate uncertainty through the non-linearity when activating hidden states with an activation function $\phi(.)$,

$$A_i^{(j+1)} = \phi\left(Z_i^{(j+1)}\right). \tag{5}$$

In order to tackle these issues, TAGI made the following assumptions

A1. The prior for the parameters and variables in the input layer are Gaussians.

A2. The product of a weight and an activation unit in Equation 4 is approximated by a Gaussian random variable, $WA \approx \mathcal{N}(\mu_{WA}, \sigma_{WA}^2)$, whose moments are computed analytically using *Gaussian multiplicative approximation* (GMA).

A3. The activation function $\phi(.)$ in Equation 5 is locally linearized using a first-order Taylor expansion evaluated at the expected value of the hidden unit being activated.

Note that the linearization procedure in the assumption A3 may seems to be a crude approximation, yet it has been shown to match or exceeds the state-of-the-art performance on fully-connected neural networks (FNN) (Goulet et al., 2021), as well as convolutional neural networks (CNN) and generative adversarial networks (Nguyen & Goulet, 2021). In order to maintain a linear computational complexity with respect to the number of weights for the forward steps, TAGI relies on the following assumptions

A4. The covariance for all parameters in the network and for all the hidden units within a same layer are diagonal.

A5. The hidden states on a given layer are independent from the parameters connecting into the subsequent layer, i.e., $\boldsymbol{Z}^{(j)} \perp\!\!\!\perp \boldsymbol{\theta}^{(j)}$.

A6. The hidden states at layer $j+1$ depend only on the hidden states and parameters directly connecting into it.

The empirical validity of those assumptions as well as the GMA formulations are provided by Goulet et al. (2021).

The second backward update-step consists in performing layer-wise recursive Bayesian inference going from hidden-layer to hidden-layer, and from hidden-layer to the parameters connecting into it (see Figure 1). The assumptions A1-A6 allow performing analytical inference while still maintaining a linear computational complexity with respect to the number of weight parameters in the network where only the posterior of hidden states for the output layer are computed using Equation 3. For the remaining layers, those quantities are obtained using the Rauch-Tung-Striebel recursive procedure that was developed in the context of state-space models (Rauch et al., 1965). For simplicity, we define the short-hand notation $\{\boldsymbol{\theta}^+, \boldsymbol{Z}^+\} \equiv \{\boldsymbol{\theta}^{(j+1)}, \boldsymbol{Z}^{(j+1)}\}$ and $\{\boldsymbol{\theta}, \boldsymbol{Z}\} \equiv \{\boldsymbol{\theta}^{(j)}, \boldsymbol{Z}^{(j)}\}$ so that the posterior for the parameters and hidden states are computed following

$$
\begin{aligned}
f(\boldsymbol{Z}|\boldsymbol{y}) &= \mathcal{N}(z; \boldsymbol{\mu}_{\boldsymbol{Z}|\boldsymbol{y}}, \boldsymbol{\Sigma}_{\boldsymbol{Z}|\boldsymbol{y}}) \\
\boldsymbol{\mu}_{\boldsymbol{Z}|\boldsymbol{y}} &= \boldsymbol{\mu}_{\boldsymbol{Z}} + \mathbf{J}_{\boldsymbol{Z}} \left(\boldsymbol{\mu}_{\boldsymbol{Z}^+|\boldsymbol{y}} - \boldsymbol{\mu}_{\boldsymbol{Z}^+}\right) \\
\boldsymbol{\Sigma}_{\boldsymbol{Z}|\boldsymbol{y}} &= \boldsymbol{\Sigma}_{\boldsymbol{Z}} + \mathbf{J}_{\boldsymbol{Z}} \left(\boldsymbol{\Sigma}_{\boldsymbol{Z}^+|\boldsymbol{y}} - \boldsymbol{\Sigma}_{\boldsymbol{Z}^+}\right) \mathbf{J}_{\boldsymbol{Z}}^{\mathsf{T}} \\
\mathbf{J}_{\boldsymbol{Z}} &= \boldsymbol{\Sigma}_{\boldsymbol{Z}\boldsymbol{Z}^+} \boldsymbol{\Sigma}_{\boldsymbol{Z}^+}^{-1},
\end{aligned}
\tag{6}
$$

$$
\begin{aligned}
f(\boldsymbol{\theta}|\boldsymbol{y}) &= \mathcal{N}(\boldsymbol{\theta}; \boldsymbol{\mu}_{\boldsymbol{\theta}|\boldsymbol{y}}, \boldsymbol{\Sigma}_{\boldsymbol{\theta}|\boldsymbol{y}}) \\
\boldsymbol{\mu}_{\boldsymbol{\theta}|\boldsymbol{y}} &= \boldsymbol{\mu}_{\boldsymbol{\theta}} + \mathbf{J}_{\boldsymbol{\theta}} \left(\boldsymbol{\mu}_{\boldsymbol{Z}^+|\boldsymbol{y}} - \boldsymbol{\mu}_{\boldsymbol{Z}^+}\right) \\
\boldsymbol{\Sigma}_{\boldsymbol{\theta}|\boldsymbol{y}} &= \boldsymbol{\Sigma}_{\boldsymbol{\theta}} + \mathbf{J}_{\boldsymbol{\theta}} \left(\boldsymbol{\Sigma}_{\boldsymbol{Z}^+|\boldsymbol{y}} - \boldsymbol{\Sigma}_{\boldsymbol{Z}^+}\right) \mathbf{J}_{\boldsymbol{\theta}}^{\mathsf{T}} \\
\mathbf{J}_{\boldsymbol{\theta}} &= \boldsymbol{\Sigma}_{\boldsymbol{\theta}\boldsymbol{Z}^+} \boldsymbol{\Sigma}_{\boldsymbol{Z}^+}^{-1}.
\end{aligned}
\tag{7}
$$

Figure 1 presents a directed acyclic graph (DAG) describing the interconnectivity in such a neural network.

TAGI allows inferring the diagonal posterior knowledge for weights and bias parameters, either using one observation at a time, or using mini-batches of data. As we will show in the next sections, this online learning capacity is best suited for RL problems where we experience episodes sequentially and where we need to define a tradeoff between exploration and exploitation, as a function of our knowledge of the expected value associated with being in a state and taking an action.

## 2.2 EXPECTED VALUE AND BELLMAN'S EQUATION

We define $r(\boldsymbol{s}, a, \boldsymbol{s}')$ as the reward for being in a state $\boldsymbol{s} \in \mathbb{R}^{\mathtt{S}}$, taking an action $a \in \mathcal{A} = \{a_1, a_2, \cdots a_{\mathtt{A}}\}$, and ending in a state $\boldsymbol{s}' \in \mathbb{R}^{\mathtt{S}}$. For simplicity, we use the short-form notation

$$\longrightarrow f(\boldsymbol{\theta}^{(j)}|\boldsymbol{y}), f(\boldsymbol{Z}^{(j)}|\boldsymbol{y})$$

$\boxed{\boldsymbol{x}}-\boldsymbol{\theta}^{(0)}\rightarrow\boxed{\boldsymbol{z}^{(1)}}-\boldsymbol{\theta}^{(1)}\rightarrow\cdots\leftarrow\boldsymbol{\theta}^{(L-1)}\boxed{\boldsymbol{z}^{(L)}}-\boldsymbol{\theta}^{(L)}\rightarrow\boxed{\boldsymbol{z}^{(0)}}\longrightarrow\boxed{\boldsymbol{y}}$

Figure 1: Compact representation of the variable nomenclature and the dependencies associated with a feedforward neural network. The red node denotes the input vector, green nodes are vectors of hidden units $\boldsymbol{z}$, and purple node denote the observation vector. The gray arrows represent the weights and bias $\boldsymbol{\theta}$ connecting the different hidden layers and magenta arrows outline the flow of information that takes place during the inference step.

for the reward $r(\boldsymbol{s}, a, \boldsymbol{s}') \equiv r(\boldsymbol{s})$ in order to define the value as the infinite sum of discounted rewards

$$v(\boldsymbol{s}) = \sum_{k=0}^{\infty} \gamma^k r(\boldsymbol{s}_{t+k}). \tag{8}$$

As we do not know what will be the future states $\boldsymbol{s}_{t+k}$ for $k > 0$, we need to consider them as random variables ($\boldsymbol{S}_{t+k}$), so that the value $V(\boldsymbol{s}_t)$ becomes a random variable as well,

$$V(\boldsymbol{s}_t) = r(\boldsymbol{s}_t) + \sum_{k=1}^{\infty} \gamma^k r(\boldsymbol{S}_{t+k}). \tag{9}$$

Rational decisions regarding which action to take among the set $\mathcal{A}$ is based the maximization of the expected value as defined by the *action-value* function

$$q(\boldsymbol{s}_t, a_t) = \mu_V \equiv \mathbb{E}[V(\boldsymbol{s}_t, a_t, \pi)] = r(\boldsymbol{s}_t) + \mathbb{E}\left[\sum_{k=1}^{\infty} \gamma^k r(\boldsymbol{S}_{t+k})\right], \tag{10}$$

where it is assumed that at each time $t$, the agent takes the action defined in the policy $\pi$. In the case of episode-based learning where the agent interacts with the environment, we assume we know the tuple of states $\boldsymbol{s}_t$ and $\boldsymbol{s}_{t+1}$, so that we can redefine the value as

$$\begin{aligned} V(\boldsymbol{s}_t, a_t) &= r(\boldsymbol{s}_t) + \gamma\left(r(\boldsymbol{s}_{t+1}) + \sum_{k=1}^{\infty} \gamma^k r(\boldsymbol{S}_{t+1+k})\right) \\ &= r(\boldsymbol{s}_t) + \gamma V(\boldsymbol{s}_{t+1}, a_{t+1}). \end{aligned} \tag{11}$$

Assuming that the value $V \sim \mathcal{N}(v; \mu_V, \sigma_V^2)$ in Equations 9 and 11 is described by Gaussian random variables, we can reparameterize these equations as the sum of the expected value $q(\boldsymbol{s}, a)$ and a zero-mean Gaussian random variable $\mathcal{E} \sim \mathcal{N}(\epsilon; 0, 1)$, so that

$$V(\boldsymbol{s}, a) = q(\boldsymbol{s}, a) + \sigma_V \mathcal{E}, \tag{12}$$

where the variance $\sigma_V^2$ and $\mathcal{E}$ are assumed here to be independent of $\boldsymbol{s}$ and $a$. Although in a more general framework this assumption could be relaxed, such an heteroscedastic variance term is outside from the scope of this paper. Using this reparameterization, we can write Equation 11 as the discounted difference between the expected values of two subsequent states

$$\begin{aligned} q(\boldsymbol{s}_t, a_t) &= r(\boldsymbol{s}_t) + \gamma q(\boldsymbol{s}_{t+1}, a_{t+1}) - \sigma_{V_t}\mathcal{E}_t + \gamma\sigma_{V_{t+1}}\mathcal{E}_{t+1} \\ &= r(\boldsymbol{s}_t) + \gamma q(\boldsymbol{s}_{t+1}, a_{t+1}) + \sigma_V \mathcal{E}. \end{aligned} \tag{13}$$

Note that in Equation 13, $\sigma_{V_t}$ and $\gamma\sigma_{V_{t+1}}$ can be combined in a single standard deviation parameters $\sigma_V$ with the assumption that $\mathcal{E}_i \perp\!\!\!\perp \mathcal{E}_j, \forall i \neq j$.

In the case where at a time $t$, we want to update the Q-values encoded in the neural net only after observing $n$-step returns (Mnih et al., 2016), we can reformulate the observation equation so that

$$q(\boldsymbol{s}_t, a_t) = \sum_{i=0}^{n-t-1} \gamma^i r(\boldsymbol{s}_{t+i}) + \gamma^{n-t} q(\boldsymbol{s}_n, a_n) + \sigma_V \mathcal{E}_t, \forall t = \{1, 2, \cdots, n-1\}. \tag{14}$$

Note that in the application of Equation 14, we employ the simplifying assumption that $\mathcal{E}_t \perp\!\!\!\perp \mathcal{E}_{t+i}, \forall i \neq 0$, as Equation 13 already makes simplifying assumptions for the independence of $\sigma_V^2$ and $\mathcal{E}$. Note that in a more general framework, this assumption could be relaxed. An example of $n$-step returns is presented in the the algorithm displayed in §1 from Appendix A.

The following subsections will present, for the case of categorical actions, how to model the deterministic action-value function $q(\boldsymbol{s}, a)$ using a neural network.

## 2.3 TAGI Deep Q-learning for Categorical Actions

Suppose we represent the environment's state at a time $t$ and $t + 1$ by $\{\boldsymbol{s}, \boldsymbol{s}'\}$, and the expected value for each of the A possible actions $a \in \mathcal{A}$ by the vector $\boldsymbol{q} \in \mathbb{R}^A$. In that context, the role of the neural network is to model the relationships between $\{\boldsymbol{s}, a\}$ and $\boldsymbol{q}$. Figure 2a presents a directed acyclic graph (DAG) describing the interconnectivity in such a neural network, where red nodes denote state variables, green nodes are vectors of hidden units $\boldsymbol{z}$, the blue box is a compact representation for the structure of a convolutional neural network, and where gray arrows represent the weights and bias $\boldsymbol{\theta}$ connecting the different hidden layers. Note that unlike other gray arrows, the magenta ones in (b) are not directed arcs representing dependencies, but they simply outline the flow of information that takes place during the inference step. For simplification purposes, the convolutional operations are omitted and all regrouped under the CNN box. In order to learn the parameters $\boldsymbol{\theta}$ of such a network,

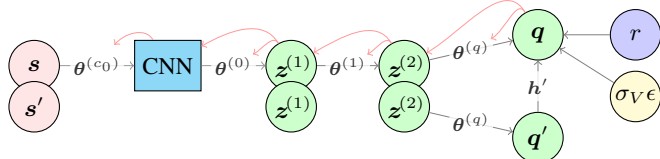

(a) Neural network DAG for modelling the action-value function $q$

(b) DAG for the temporal-difference Q-learning configuration

Figure 2: Graphical representation of a neural network structure for temporal-difference Q-learning with categorical actions. The red nodes denote state variables, green nodes are vectors of hidden units $\boldsymbol{z}$, and the blue box is a compact representation for the structure of a convolutional neural network. The gray arrows represent the weights and bias $\boldsymbol{\theta}$ connecting the different hidden layers and magenta arrows outline the flow of information that takes place during the inference step.

we need to expand the graph from Figure 2a to include the reward $r$, the error term $\sigma_V \epsilon$, and $\boldsymbol{q}'$, the $q$-values of the time step $t + 1$. This configuration is presented in Figure 2b where the nodes that have been doubled represent the states $\boldsymbol{s}$ and $\boldsymbol{s}'$ which are both evaluated in a network sharing the same parameters. Note that the value of $\boldsymbol{q}'$ is computed using the same network presented in Figure 2a in which this network acts as a predictor taking input as $\boldsymbol{s}'$. When applying Equation 13, $q$-values corresponding to a specific action can be selected using a vector $\boldsymbol{h}_i \in \{0, 1\}^A$ having a single non-zero value for the $i$-th component identifying which action was taken at a time $t$ so that

$$q_i = [\boldsymbol{q}]_i = \boldsymbol{h}_i^\mathsf{T} \boldsymbol{q}. \tag{15}$$

In the context TAGI-DQN, we employ Thompson sampling (Strens, 2000) for the selection of an action given a state. More specifically, the vector $\boldsymbol{h}_i' \in \{0, 1\}^A$ is defined such that the $i$-th non-zero value corresponds to the index of the maximal value among $\boldsymbol{q}'$, a vector of realizations from the neural network's posterior predictive output $\boldsymbol{Q} \sim \mathcal{N}(\boldsymbol{q}'; \boldsymbol{\mu}_{\boldsymbol{Q}|\mathcal{D}}, \boldsymbol{\Sigma}_{\boldsymbol{Q}|\mathcal{D}})$. Because of the Gaussian assumptions in TAGI, this posterior predictive is readily available from the forward uncertainty propagation step, as outlined in §2.1.

The red arrows in Figure 2b outline the flow of information during the inference procedure. The first step consists in inferring $\boldsymbol{q}$ using the relationships defined in either Equation 13 or 14. As this is a linear equation involving Gaussian random variables, the inference is analytically tractable. From there, one can follow the same layer-wise recursive procedure presented in §2.1 in order to learn

the weights and biases in $\boldsymbol{\theta}$. With the exclusion of the standard hyperparameters related to network architecture, batch size, buffer size or the discount factor, this TAGI-DQN framework only involves a single hyperparameter, $\sigma_V$, the standard deviation for the value function. Note that when using CNNs with TAGI, Nguyen & Goulet (2021) recommended using a decay function for the standard deviation of the observation noise so that after seing $e$ batches of $n$-steps,

$$\sigma_V^e = \max(\sigma_V^{\min}, \eta \cdot \sigma_V)^{e-1}, \tag{16}$$

because it allows putting more weight on the prior rather than on the likelihood during the inference step at the early stage. This effect then diminishes with time. The model in Equation 16 has three hyperparameters, the minimal noise parameter $\sigma_V^{\min}$, the decay factor $\eta$ and the initial noise parameter $\sigma_V$. As it was shown by Nguyen & Goulet (2021) for CNNs and how we show in §4 for RL problems, TAGI's performance is robust towards the selection of these hyperparameters.

A comparison of implementation between TAGI and backpropagation on deep Q-network with experience replay (Mnih et al., 2015) is shown in Figure 3. A practical implementation of $n$-step TAGI deep Q-learning is presented in Algorithm 1 from Appendix A

---

**Algorithm 1:** TAGI-DQN with Experience Replay

1  Initialize replay memory $\mathcal{R}$ to capacity $N$; $\boldsymbol{\Sigma}_V$;
2  Initialize parameters $\boldsymbol{\theta}$;
3  Discount factor $\gamma$;
4  **for** *episode* $= 1 : \mathtt{E}$ **do**
5      Reset environment $\mathbf{s}_0$;
6      **for** $t = 1 : \mathtt{T}$ **do**
7          $q(s_t, a) : Q(s_t, a) \sim \mathcal{N}(\boldsymbol{\mu}_{\boldsymbol{\theta}}^Q(s_t, a), \boldsymbol{\Sigma}_{\boldsymbol{\theta}}^Q(s_t, a))$;
8          $a_t = \arg\max\limits_{a \in \mathcal{A}} q(s_t, a)$;
9          $s_{t+1}, r_t = $ enviroment$(a_t)$;
10         Store $\{s_t, a_t, r_t, s_{t+1}\}$ in $\mathcal{R}$;
11         Sample random batch of $\{s_j, a_j, r_j, s_{j+1}\}$;
12         $q(s_{j+1}, a') : Q(s_{j+1}, a') \sim \mathcal{N}(\boldsymbol{\mu}_{\boldsymbol{\theta}}^Q(s_{j+1}, a'), \boldsymbol{\Sigma}_{\boldsymbol{\theta}}^Q(s_{j+1}, a'))$;
13         $a'_{j+1} = \arg\max\limits_{a' \in \mathcal{A}} q(s_{j+1}, a')$;
14         $\boldsymbol{\mu}_{Y_j} = r_j + \gamma \boldsymbol{\mu}_{\boldsymbol{\theta}}^Q(s_{j+1}, a'_{j+1})$;
15         $\boldsymbol{\Sigma}_{Y_j} = \gamma^2 \boldsymbol{\Sigma}_{\boldsymbol{\theta}}^Q(s_{j+1}, a'_{j+1}) + \boldsymbol{\Sigma}_V$;
16         Update $\boldsymbol{\theta}$ using TAGI on $f(\boldsymbol{\theta}|\mathbf{y})$ (Equation 7);

**Algorithm 2:** DQN with Experience Replay

1  Initialize replay memory $\mathcal{R}$ to capacity $N$;
2  Initialize parameters $\boldsymbol{\theta}$;
3  Discount factor $\gamma$;
4  Define $\epsilon$ (epsilon-greedy function);
5  **for** *episode* $= 1 : \mathtt{E}$ **do**
6      Reset environment $\mathbf{s}_0$;
7      **for** $t = 1 : \mathtt{T}$ **do**
8          $u : U \sim \mathcal{U}(0, 1)$;
9          $a_t = \begin{cases} \mathtt{randi}(\mathtt{A}) & u < \epsilon; \\ \arg\max\limits_{a \in \mathcal{A}} Q_{\boldsymbol{\theta}}(s_t, a) & u \geq \epsilon; \end{cases}$
10         $s_{t+1}, r_t = $ enviroment$(a_t)$;
11         Store $\{s_t, a_t, r_t, s_{t+1}\}$ in $\mathcal{R}$;
12         Sample random batch of $\{s_j, a_j, r_j, s_{j+1}\}$;
13         $y_j = r_j + \gamma \max\limits_{a' \in \mathcal{A}} Q_{\boldsymbol{\theta}}(s_{j+1}, a')$;
14         Update $\boldsymbol{\theta}$ using gradient descent on
15         $L = 0.5 \left[ y_j - Q_{\boldsymbol{\theta}}(s_j, a_j) \right]^2$;

Figure 3: Comparison of TAGI with backpropagation on deep Q-network with experience replay. $L$: loss function; $\mathcal{U}$: uniform distribution; $\mathtt{randi}$: uniformly distributed pseudorandom integers.

## 3  RELATED WORKS

Over the last decades, several approximate methods have been proposed in order to allow for Bayesian neural networks (Neal, 1995; Kingma et al., 2015; Hernández-Lobato & Adams, 2015; Blundell et al., 2015; Louizos & Welling, 2016; Osawa et al., 2019; Wu et al., 2019; Gal & Ghahramani, 2016) with various degree of approximations. Although some these methods have shown to be capable of tackling classification tasks on datasets such ImageNet (Osawa et al., 2019), none of them have been applied on large-scale RL benchmark problems such as the Atari games environement. The key idea behind using Bayesian methods for reinforcement learning is to consider the uncertainty associated with Q-functions in order to identify a tradeoff between exploring the performance of possible actions and exploiting the current optimal policy (Sutton & Barto, 2018). This typically takes the form of performing Thompson sampling (Strens, 2000) rather than relying on heuristics such as $\epsilon$-greedy.

For instance, MC dropout (Gal & Ghahramani, 2016) was introduced has a method intrinsically suited for reinforcement learning. Nevertheless, five years after its inception, the approach has not yet been reliably scaled to more advanced benchmarks such as the Atari game environment. The

same applies to Bayes-by-backprop (Blundell et al., 2015) which was recently applied to simple RL problems (Lipton et al., 2018), and which has not yet been applied to more challenging environments requiring convolutional networks. On the other hand, Bayesian neural networks relying on sampling methods such as Hamiltonian Monte-Carlo (Neal, 1995) are typically computationally demanding to be scaled to RL problems involving such a complex environment.

Although mainstream methods related to Bayesian neural networks have seldom been applied to complex RL problems, several research teams have worked on alternative approaches in order to allow performing Thompson sampling. For instance, Azizzadenesheli et al. (2018) have employed a deep Q-network where the output layer relies on Bayesian linear regression. This approach was shown to be outperforming its deterministic counterparts on Atari games. Another approach by Osband et al. (2016) employs bootstrapped deep Q-networks with multiple network heads in order to represent the uncertainty in the Q-functions. This approach was also shown to scale to Atari games while presenting an improved performance in comparison with deterministic deep Q-networks. Finally, Wang & Zhou (2020) have tackled the same problem, but this time by modelling the variability in the Q-functions through a latent space learned using variational inference. Despite its good performance on the benchmarks tested, it did not allowed to be scaled to the Atari game environment.

The TAGI deep Q-network presented in this paper is the first demonstration that an analytically tractable inference approach for Bayesian neural networks can be scaled to a problem as challenging as the Atari game environment.

## 4 BENCHMARKS

This section compares the performance of TAGI with backpropagation-based standard implementations on off- and on-policy deep RL. For the off-policy RL, both TAGI-based and backpropagation-based RL approaches are applied to deep Q-learning with experience replay (see Algorithm 1 & 2) for the lunar lander and cart-pole environments. For the on-policy RL, TAGI is applied to the $n$-step Q-learning algorithm (Mnih et al., 2016) and is compared with its backpropagation-based counterpart. For this purpose, we perform the comparison on five Atari games including Beamrider, Breakout, Pong, Qbert, and Space Invaders. Note that these five games are commonly selected for tuning hyperparameters for the entire Atari games (Mnih et al., 2016; 2013) because they represent the features sharing with the remaining Atari games. All benchmark environments are taken from the OpenAI Gym (Brockman et al., 2016).

### 4.1 EXPERIMENTAL SETUP

In the first experiments with off-policy RL, we use a fully-connected multilayer perceptron (MLP) with two hidden layers of 256 units for the lunar lander environment, and with one hidden layer of 64 units for the cart pole environment. In these experiments, there is no need for input processing nor for reward normalization. Note that unlike for the deterministic deep Q-network with experience replay, TAGI does not separate the Q-network with the target network that is used for ensuring the stability during training and allows eliminating the hyperparameter related to the target update frequency. For the deep Q-network trained with backpropagation, we employ the pre-tuned implementation of OpenAI baselines (Dhariwal et al., 2017) with all hyperparameters set to the default values.

For the Atari experiments with on-policy RL, we use the same input processing and model architecture as Mnih et al. (2016). More specifically, the Q-network uses two convolutional layers (16-32 filters) and a full-connected MLP of 256 units. The $n$-step Q-learning only uses a single network to represent the value function for each action, and relies on a single learning agent in this experiment. The reason behind this choice is that TAGI current main library is only available on Matlab which does not support running a Python multiprocessing module such as the OpenAI gym. In the context of TAGI, we use an horizon of 128 steps. As recommended by Andrychowicz et al. (2021) and following practical implementation details (Pytorch, 2019; 2020), each return in $n$-step Q-learning algorithm is normalized by subtracting the average return from the current $n$-steps and then dividing by the empirical standard deviation from the set of $n$ returns.

The standard deviation for the value function, $(\sigma_V)$, is initialized at 2. $\sigma_V$ is decayed each 128 steps with a factor $\eta = 0.9999$. The minimal standard deviation for the value function $\sigma_V^{\min} = 0.3$. These hyperparameters values were not grid-searched but simply adapted to the scale of the problems

and are kept constant for both experiments. The complete details of the network architecture and hyperparameters are provided in Appendix B and C.

## 4.2 RESULTS

For the first set of experiments using off-policy RL, Figure 4 presents the average reward over 100 episodes for three runs for the lunar lander and cart pole environment. The TAGI-based deep Q-learning with experience replay shows a faster and more stable learning than the one relying on backpropagation.

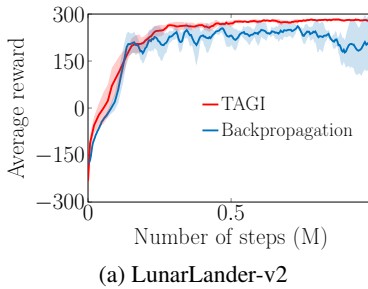
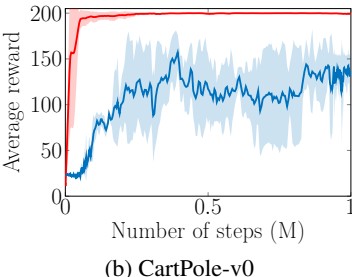

(a) LunarLander-v2                          (b) CartPole-v0

Figure 4: Illustration of average rewards over 100 episodes of five runs for one million time steps for the TAGI-based and backpropagation-based deep Q-learning.

Table 1 shows that the average reward over the last 100 episodes obtained using TAGI are greater than the one obtained using backpropagation. Figure 5 compares the average reward over 100 episodes

Table 1: Average reward over the last 100 episodes for the lunar lander and cart pole experiments. TAGI: Tractable Approximate Gaussian Inference.

| Method | Lunar lander | Cart pole |
|---|---|---|
| TAGI | $279.9 \pm 6.1$ | $199.3 \pm 1.0$ |
| Backpropagation | $197.2 \pm 84.3$ | $137.7 \pm 16.5$ |

for three runs obtained for TAGI, with the results from Mnih et al. (2016) for the second set of experiments on Atari games. Note that all results presented were obtained for a single agent, and that the results for the backpropagation-trained networks are only reported at the end of each epoch.

Results show that TAGI outperforms the results from the original $n$-step Q-learning algorithm trained with backpropagation (Mnih et al., 2016) on Breakout, Pong, and Qbert, while underperforming on Beam Rider and Space Invaders. The average training time of TAGI for an Atari game is approximately 13 hours on GPU calculations benchmarked on a 4-core-intel desktop of 32 GB of RAM with a NVIDIA GTX 1080 Ti GPU. The training speed of TAGI for the experiment of the off-policy deep RL is approximately three times slower on CPU calculations than the backpropagation-trained counterpart. The reason behind this slower training time is because of its intrinsically different inference engine making TAGI's implementation incompatible with existing libraries such as TensorFlow or Pytorch. TAGI's library development is still ongoing and it is not yet fully optimized for computational efficiency. Overall, these results for on- and off policy RL approaches confirm that TAGI can be applied to large scale problems such as deep Q-learning.

## 5 DISCUSSION

Although the performance of TAGI does not systematically outperform its backpropagation-based counterpart, it requires fewer hyperparameters (see Appendix C). This advantage is one of the key

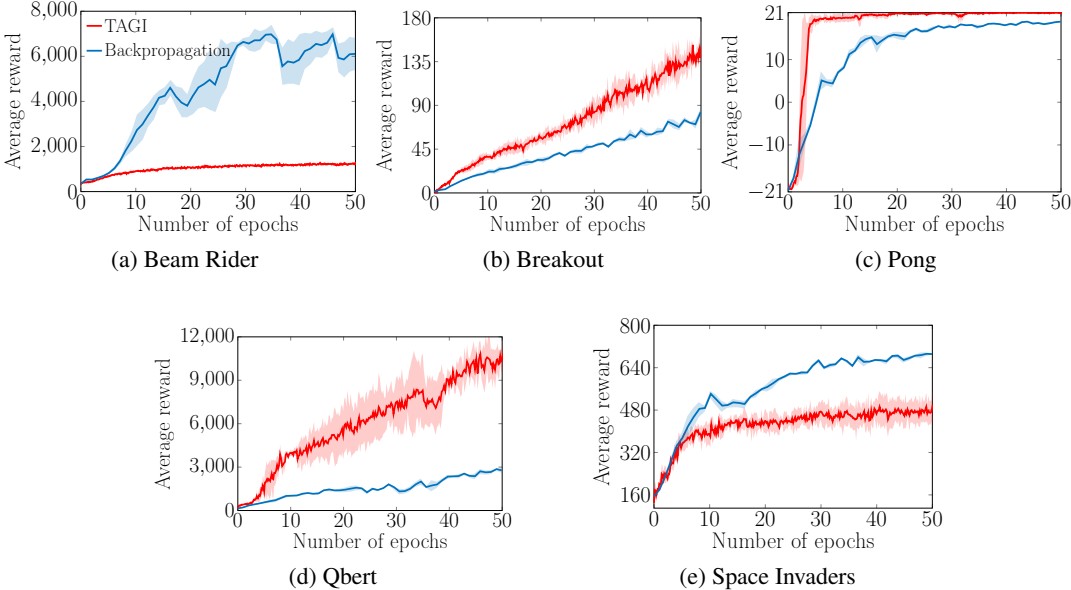

Figure 5: Illustration of average reward over 100 episodes of three runs for five Atari games. The number of epochs is used here for the comparison of TAGI and backpropagation-trained counterpart obtained by Mnih et al. (2016). Each epoch corresponds to four million frames. The environment identity are {*Atari Game*}`NoFrameSkip-v4`.

aspects for improving the generalization and reducing the computational cost of the hyperparameter tuning process which are the key challenges in current state of deep RL (Irpan, 2018; Henderson et al., 2018). For instance, in this paper, the TAGI's hyperparameters relating to the standard deviation of value function ($\sigma_V$) are kept constant across all experiments. Moreover, since these hyperparameters were not subject to grid-search in order to optimize the performance, the results obtained here are representative of what a user should obtain by simply adapting the hyperparameters to fit the specificities and scale of the environment at hand.

More advanced RL approaches such as advanced actor critic (A2C) (Mnih et al., 2016) and proximal policy optimization (PPO) (Schulman et al., 2017) employ two-networks architectures in which one network is used to approximate a value function and other is employed to encode the policy. The current TAGI-RL framework is not yet able to handle such architectures because training a policy network involves an optimization problem for the selection of the optimal action. Backpropagation-based approach currently rely on gradient optimization to perform this task, while TAGI will require developing alternative approaches in order to maintain the analytical tractability without relying on gradient-based optimization.

## 6  CONCLUSION

This paper proposes a Bayesian reinforcement learning framework that combines TAGI with deep Q-learning. The proposed framework allows estimating the uncertainty of neural network's parameters and taking an action given a state according to the parameter uncertainty using Thompson sampling. Throughout the experiments with on- and off-policy reinforcement learning approaches, we demonstrate that TAGI can reach a performance comparable to backpropagation-trained networks while using only half the number of hyperparameters. These results challenge the common belief that for large scale problems such as the Atari environment, neural networks can only be trained using the gradient backpropagation. We have shown here that this current paradigm is no longer the only alternative as TAGI has a linear computational complexity and can be used to learn the parameters of complex networks in an analytically tractable manner, without relying on gradient-based optimization.

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

# A  ALGORITHM

This section presents the $n$-steps Q-learning algorithm with Tractable Approximate Gaussian Inference (TAGI).

---
**Algorithm 3:** $n$-step Q-learning with TAGI

---
1  Initialize $\boldsymbol{\theta}$ ; $\boldsymbol{\Sigma}_V$; number of steps $(N)$
2  Initialize memory $\mathcal{R}$ to capacity $N$;
3  steps = 0;
4  **for** *episode* $= 1 : $ E **do**
5      Reset environment $\mathbf{s}_0$;
6      **for** $t = 1 : $ T **do**
7         steps = steps + 1;
8         $q(s_t, a) : Q(s_t, a) \sim \mathcal{N}(\boldsymbol{\mu}_{\boldsymbol{\theta}}^Q(s_t, a), \boldsymbol{\Sigma}_{\boldsymbol{\theta}}^Q(s_t, a))$;
9         $a_t = \underset{a \in \mathcal{A}}{\arg\max}\, q(s_t, a)$;
10        $s_{t+1}, r_t = $ enviroment$(a_t)$;
11        Store $\{s_t, a_t, r_r\}$ in $\mathcal{R}$;
12        **if** *steps mod* $N == 0$ **then**
13           $q(s_{t+1}, a') : Q(s_{t+1}, a') \sim \mathcal{N}(\boldsymbol{\mu}_{\boldsymbol{\theta}}^Q(s_{t+1}, a'), \boldsymbol{\Sigma}_{\boldsymbol{\theta}}^Q(s_{t+1}, a'))$;
14           $a'_{t+1} = \underset{a \in \mathcal{A}}{\arg\max}\, q(s_{t+1}, a')$;
15           Take $N$ samples of $\{s_j, a_j, r_j\}$ from $\mathcal{R}$;
16           $\mu_N^y = \boldsymbol{\mu}_{\boldsymbol{\theta}}^Q(s_{t+1}, a'_{t+1}); \Sigma_N^y = \boldsymbol{\Sigma}_{\boldsymbol{\theta}}^Q(s_{t+1}, a'_{t+1})$;
17           **for** $j = N - 1 : 1$ **do**
18              $\mu_j^y = r_j + \gamma \mu_{j+1}^y; \Sigma_j^y = \gamma^2 \Sigma_{j+1}^y + \Sigma_V$;
19           Update $\boldsymbol{\theta}$ using TAGI;
20           Initialize memory $\mathcal{R}$ to capacity $N$;

---

# B  MODEL ARCHITECTURE

This appendix contains the specifications for each model architecture in the experiment section. $D$ refers to a layer depth; $W$ refers to a layer width; $H$ refers to the layer height in case of convolutional or pooling layers; $K$ refers to the kernel size; $P$ refers to the convolutional kernel padding; $S$ refers to the convolution stride; $\sigma$ refers to the activation function type; ReLU refers to rectified linear unit; $N_a$ refers to the number of actions.

Table 2: Model Architecture for Cart pole

| Layer | $D \times W \times H$ | $K \times K$ | $P$ | S | $\sigma$ |
|---|---|---|---|---|---|
| Input | $4 \times 1 \times 1$ | - | - | - | - |
| Full connected | $64 \times 1 \times 1$ | - | - | - | ReLU |
| Output | $2 \times 1 \times 1$ | - | - | - | - |

# C  HYPERPARAMETER

This appendix details the hyperparameters for each model architecture in the experiment section

Table 3: Model Architecture for Lunar lander

| Layer | $D \times W \times H$ | $K \times K$ | $P$ | S | $\sigma$ |
|---|---|---|---|---|---|
| Input | $8 \times 1 \times 1$ | - | - | - | - |
| Full connected | $256 \times 1 \times 1$ | - | - | - | ReLU |
| Full connected | $256 \times 1 \times 1$ | - | - | - | ReLU |
| Output | $4 \times 1 \times 1$ | - | - | - | - |

Table 4: Model Architecture for Atari domain

| Layer | $D \times W \times H$ | $K \times K$ | $P$ | S | $\sigma$ |
|---|---|---|---|---|---|
| Input | $4 \times 84 \times 84$ | - | - | - | - |
| Convolutional | $16 \times 20 \times 20$ | $8 \times 8$ | 0 | 4 | ReLU |
| Convolutional | $32 \times 9 \times 9$ | $4 \times 4$ | 0 | 2 | ReLU |
| Full connected | $256 \times 1 \times 1$ | - | - | - | ReLU |
| Output | $N_a \times 1 \times 1$ | - | - | - | |

Table 5: Hyperparameters for Cart pole and Lunar lander

| Method | # | Hyperparameter | Value |
|---|---|---|---|
| TAGI | 1 | Initial standard deviation for the value function ($\sigma_V$) | 2 |
| | 2 | Decay factor ($\eta$) | 0.9999 |
| | 3 | Minimal standard deviation for the value function ($\sigma_V^{\min}$) | 0.3 |
| | 4 | Buffer size | 50 000 |
| | 5 | Batch size | 10 |
| | 6 | Discount ($\gamma$) | 0.99 |
| Backprop | 1 | Learning rate | $5 \times 10^{-4}$ |
| | 2 | Adam epsilon | $10^{-5}$ |
| | 3 | Adam $\beta_1$ | 0.9 |
| | 4 | Adam $\beta_2$ | 0.999 |
| | 5 | Buffer size | 50 000 |
| | 6 | Exploration fraction | 0.1 |
| | 7 | Final value of random action probability | 0.02 |
| | 8 | Batch Size | 32 |
| | 9 | Discount ($\gamma$) | 0.99 |
| | 10 | Target update frequency | 500 |
| | 11 | Gradient norm clipping coefficient | 10 |

Table 6: Hyperparameters for Atari domain

| Method | # | Hyperparameter | Value |
|---|---|---|---|
| TAGI | 1 | Horizon | 128 |
| | 2 | Initial standard deviation for the value function ($\sigma_V$) | 2 |
| | 3 | Decay factor ($\eta$) | 0.9999 |
| | 4 | Minimal standard deviation for the value function ($\sigma_V^{\min}$) | 0.3 |
| | 5 | Batch size | 32 |
| | 6 | Discount ($\gamma$) | 0.99 |
| | 7 | Number of actor-learners | 1 |
| Backprop | 1 | Horizon | 5 |
| | 2 | Initial learning rate | $LogUniform(10^{-4}, 10^{-2})$ |
| | 3 | Learning rate schedule | LinearAnneal$(1, 0)$ |
| | 4 | RMSProp decay parameter | 0.99 |
| | 5 | Exploration rate 1 ($\epsilon_1$) | 0.1 |
| | 6 | Exploration rate 2 ($\epsilon_2$) | 0.01 |
| | 7 | Exploration rate 3 ($\epsilon_3$) | 0.5 |
| | 8 | Probability of exploration rate 1 | 0.4 |
| | 9 | Probability of exploration rate 2 | 0.3 |
| | 10 | Probability of exploration rate 3 | 0.3 |
| | 11 | Exploration rate schedule (first four million frames) | Anneal from 1 to $\epsilon_1, \epsilon_2, \epsilon_3$ |
| | 12 | Batch size | 5 |
| | 13 | Discount ($\gamma$) | 0.99 |
| | 14 | Number of actor-learners | 1 |

