# OpenReview forum: "Analytically Tractable Bayesian Deep Q-Learning"
_ICLR.cc/2022/Conference — ICLR 2022 Submitted_

### Official Review · Reviewer_Aorz · 2021-11-01

**Correctness:** 2
**Technical Novelty And Significance:** 3
**Empirical Novelty And Significance:** 2
**Recommendation:** 3
**Confidence:** 3

**Main Review:**

### Strengths:
1. Potentially useful to be able to implement deep Q-learning with the tractable approximate Gaussian inference (TAGI) method, rather than with the standard semi-gradient approach.
2. Demonstrates an alternative implementation of deep Q-learning at the scale of Atari games.
3. Proposes a fully Bayesian way of accounting for neural network parameter uncertainty in the value function during action selection.
4. Offers a strategy for reducing the number of hyperparameters involved in deep Q-learning.

----

### Weaknesses:

This paper does not focus on a clear contribution. The abstract claims that using Bayesian methods to account for the uncertainty of the NN’s parameters is key to improving decision making, but the evaluation focuses only on reducing hyperparameters and obtaining comparable performance to existing methods. Each of the strengths listed in the section above could be a fine topic for a paper if executed well, but the submitted paper moves on before fully answering any of the questions it raises.

(Note: each of the numbered weaknesses below corresponds with the same numbered strength above.)

1. If the objective is to show that it is both possible and useful to implement deep Q-learning with TAGI, then I would expect to see a full derivation for _Q-learning_, as well as some motivation or evidence for  _why_ it is useful.
    - Currently the paper only appears to consider the derivation for _value estimation_. My (somewhat uninformed) impression of the original TAGI paper is that it demonstrated both regression and classification in deep neural networks. So if one were to take the perspective that estimating the value function amounts to solving a regression problem, then the original paper already demonstrated, at least in principle, that this was possible.
    - But the value estimation problem is only one aspect of Q-learning, and it’s not obvious from the derivation how actions would fit in. Eqn (10) defines $q(s_t,a_t)$ in terms of a fixed policy, which does not make the dependence on actions clear. Then the paper redefines $V(s_t, a_t)$ as being action-dependent, which seems like an error, without describing what this dependence looks like (eqn. (11)).
    - Finally, the paper does not provide much motivation for _why_ this alternative implementation would be desirable.
2. If the objective is to demonstrate that TAGI can support deep Q-learning at scale, then I would expect an answer to the question: “how well does it work?”. I feel like the paper provides a partial answer, but the experiments don’t give me enough confidence to answer the question decisively.
    - For instance, the results are only for 3 seeds, and the shaded regions are not labeled in the plots, so it’s unclear whether the results are significant. In general, my expectation with Atari would be to see at least 5 seeds, and ideally 10 or more. With CartPole and Lunar Lander, experiments are typically much cheaper to run, and I would expect to see more like 100+ seeds. See Henderson et al. (2018) for a discussion of why this is important.
    - It’s unclear what the baseline agent is for the Atari experiments. Is it standard DQN? If so, I’m surprised by the Atari baseline performance curves. Performance is much worse than the original for Breakout (\~150 here, vs. 401.2), Qbert (\~2900 vs. 10596), and Space Invaders (\~450 vs. 1976). After reading through the appendix, my impression is that the “Backpropagation” agent trains on the same amount of experience as the DQN from Mnih et al. (2015)’s Nature paper, but the hyperparameters and network architecture seem to be different. I’m left wondering why the paper uses these settings rather than comparing against the original published DQN.
3. Given that the method claims to offer a tractable way to account for NN parameter uncertainty, I would expect to see a clear indication that this enables an agent to make _better_ decisions than methods that either don’t take a fully Bayesian approach or that don’t account for such uncertainty at all. Instead, the paper only claims to achieve “comparable” performance, and only compares against one baseline. I feel that there are a few problems with this.
    - In my opinion, “comparable” doesn’t accurately characterize the performance. I would instead say the results are “mixed” versus the “backpropagation” agent. On the Atari games, TAGI appears to be better on three out of five, but much worse on one and somewhat worse on another.
    - Regardless, neither “comparable” nor “mixed” results clearly demonstrate _better_ decision making capabilities.
    - If the story is in fact more complicated---i.e. if properly accounting for parameter uncertainty doesn’t _always_ lead to better performance---then I would want to understand when to expect it to be better and when to expect it to be worse. The experiments suggest that this may be the case, but such a discussion is absent from the results presented in the paper.
    - The paper cites Azizzadenesheli et al. (2018) and Osband et al. (2016), both of which appear to account for uncertainty and allow for Thompson sampling during action selection. I’m left wondering why the paper doesn’t compare against those approaches when evaluating the performance of the proposed method.
4. I am unconvinced that the paper demonstrates that TAGI reduces the number of hyperparameters required for deep Q-learning. Examining the hyperparameters in the appendix, I don’t find the comparison conclusive because the decisions about what should count as a hyperparameter are subjective.
    - For example, half of the “backprop” agent’s hyperparameters are related to the epsilon schedule, whereas the same agent’s learning rate schedule is represented as a single hyperparameter. Alternate forms of epsilon schedule exist that could reduce this number significantly. The paper does not justify why this particular way of accounting for them is appropriate.
    - If the objective is simply to reduce the number of hyperparameters, are there other strategies worth comparing against? What makes this strategy for reducing hyperparameters so compelling? Is there a way to go fully Bayesian and incorporate the hyperparameters into the graph, so as to effectively get rid of _all_ of them?

----

### Questions:
1. Can you comment on why it is reasonable to assume that the value V is well approximated by a Gaussian distribution? Can we expect this assumption to hold in general, or only for specific environments?
2. Is the backprop baseline intended to exactly reproduce the Nature DQN result? If so, why are the results so different from the original paper? If not, what was the rationale for using the particular DQN architecture and hyperparameters for the Atari experiments?
3. Any idea why TAGI performs better than backprop on some Atari games but worse on others?
4. Was there a reason for not comparing against Azizzadenesheli et al. (2018) and Osband et al. (2016)? Is there a benefit to the proposed method over these other approaches, aside from mathematical rigor?
5. Why does TAGI use a horizon of 128 steps, but the backpropagation baseline uses a horizon of 5 steps? Is this a fair comparison?
6. I’m confused about the exploration rates for backprop on Atari. How are the different epsilon values used in the schedule?
7. What does it mean for the initial learning rate to be LogUniform(1e-4, 1e-2)? Does this mean the learning rate is chosen randomly for each seed?
8. The exploration fraction (0.1)---which I assume is the same thing as epsilon---for backprop on CartPole and Lunar Lander seems rather high. Are you decaying that to 0.02? If not, how was this value chosen?
9. It’s interesting that the variance for TAGI seems to be much lower than for backprop on CartPole and Lunar Lander, but much higher on Atari. Does this hold up with more seeds? What might explain this behavior?
10. Is there a potential connection between entropy regularized RL and this notion of using a decay function for the observation noise variance to more strongly incentivize the prior during early learning?

----

### Comments:
1. For a reader unfamiliar with TAGI, it’s unclear which parts of this paper are reviewing existing work and which are proposing original contributions.
2. In section 2.1, $J_Z$ and $J_\theta$ are undefined. I assume these refer to the Jacobian? It would be useful to say what they are explicitly.
3. In section 2.2, the “short form notation for the reward” makes it difficult to see how actions fit into equations (8)-(11).
4. Final paragraph of Sec. 5: “advanced actor critic” should be “advantage actor critic”. I also don’t think Mnih et al. (2016) is the right reference for A2C, since that paper only discusses the asynchronous version, A3C. OpenAI Baselines or “Learning to Reinforcement Learn” by Wang et al. (2016) would be more appropriate.
5. The formatting of the references is inconsistent. Some references use the full name of the publication, while some use abbreviations. Some include conference paper URLs; others don’t, etc. References should consistently use the same style, and use the full publication name (with proper capitalization) rather than the abbreviation.

----

### References:
1. **Henderson et al. (2018) –** ​​Peter Henderson, Riashat Islam, Philip Bachman, Joelle Pineau, Doina Precup, and David Meger. "Deep Reinforcement Learning that Matters." In Proceedings of the Thirty-Second AAAI Conference on Artificial Intelligence, pp. 3207--3214, 2018.
2. **OpenAI Baselines –** either https://github.com/openai/baselines or https://arxiv.org/abs/1708.05144
3. **Wang et al. (2016) –** https://arxiv.org/abs/1611.05763



**Summary Of The Paper:**

The paper describes how to adapt the Q-learning algorithm so that it is compatible with Bayesian deep learning, as opposed to the standard (semi-)gradient-based deep learning implementation. It explains how to propagate uncertainties about network parameters through the neural network computations and ultimately to the q-values, which enables action selection using Thompson sampling. Finally, it presents empirical results on CartPole, Lunar Lander, and five Atari games.

**Summary Of The Review:**

The paper explores using Bayesian methods to guide action selection in deep Q-learning, and it makes some progress in that direction. I appreciate the time and effort spent in preparing this submission, and I do see some value in the proposed approach. Unfortunately, in its current form, the paper does not have a clear contribution. The paper attempts to answer too many questions and does not sufficiently answer any of them. Additionally, the main derivation is hard to follow and appears to be incomplete or incorrect. I recommend rejection.

---

> ### Author Response · Authors · 2021-11-16
> **Rebuttal and Clarificaitions**
>
> In a reinforcement learning problem, an agent selects an action that is based on its previous interactions with the environment and that maximizes the cumulative rewards.  This is known as the greedy policy. However, this approach fails when the rewards are uncertain. In order to address this problem, the current form of deep Q-learning method introduces occasional random actions or some other forms of enhancing exploration i.e. the tradeoff between the exploitation and exploration.  According  to Strens (2000),  a theoretically justifiable approach is (1) to retain a notion of uncertainty in the model parameters and, (2) to take decisions according to hypotheses for the true model parameters. The current form of backpropagation is unable to model the uncertainty of a neural network that approximates the value function.
>
> The main objective of our paper proposes a new framework that couples the temporal difference Q-learning with the Tractable Approximate Gaussian Inference (TAGI).  More specifically, TAGI provides the estimation of the model parameter uncertainty using closed-form Bayesian inference, and the Thompson sampling then factors the model parameter uncertainty into the selection of an action. In addition, the proposed framework can potentially address two major practical issues (Irpan 2018; Henderson et al. 2018):
> 1) automatic balances between the exploration and exploitation
> 2) number of hyperparameters to be tuned on the same algorithms being compared with our approach.
>
> The potential of the proposed framework is then illustrated on off-and on policy RL approaches with 7 different environments (Cart pole, Lunar lander, five Atari games). Note that these five Atari games are commonly selected to tune the hyperparameters related to the model and the exploitation/exploration procedure for the rest of Atari games (Mnih et al., 2016; 2013)
>
> Our intention is not to show the superiority of TAGI at any of the applications presented. Instead, we want to share with the AI/ML community a new perspective on probabilistic inference for Bayesian deep reinforcement learning which, up to now, has been relying on gradient-based methods. That being said, it does not mean that the gradient-based methods are bad, but we should be open to alternatives.   We do not think that the algorithm presented here is a finality, but only a starting point. In our opinion, this is what the research is meant to be.
>
> References:
>
> Strens 2000: A Bayesian Framework for Reinforcement Learning. https://www.ece.uvic.ca/~bctill/papers/learning/Strens_2000.pdf
>
> Henderson et al. 2018: Deep Reinforcement Learning that Matters. https://arxiv.org/pdf/1709.06560.pdf
>
> Irpan 2018: Deep reinforcement learning doesn’t work yet. https://www.alexirpan.com/2018/02/14/rl-hard.html
>
> Mnih et al. 2016: Asynchronous Methods for Deep Reinforcement Learning. https://arxiv.org/pdf/1602.01783.pdf
>
> Mnih et al. 2013: Playing atari with deep reinforcement learning https://arxiv.org/pdf/1312.5602.pdf

---

> > ### Author Response · Authors · 2021-11-16
> > **Regarding your questions**
> >
> > **Q1: Can you comment on why it is reasonable to assume that the value V is well approximated by a Gaussian distribution? Can we expect this assumption to hold in general, or only for specific environments?**
> >
> > Reply: From our test on discrete and continuous-action RL, the Gaussian assumption works well in all cases tested.
> >
> > **Q2: Is the backprop baseline intended to exactly reproduce the Nature DQN result? If so, why are the results so different from the original paper? If not, what was the rationale for using the particular DQN architecture and hyperparameters for the Atari experiments?**
> >
> > Reply: No, we compared TAGI with the n-step Q-learning algorithm,  i.e., on-policy reinforcement learning ( Figure 3, Mnih et al., 2016). We employed a single agent for this comparative study. The reason behind this choice is that TAGI current main library is only available on Matlab which does not support running a Python multiprocessing module such as the OpenAI gym. All of these details have been mentioned in section 4.1 in the manuscript.
> >
> > References:
> >
> > Mnih et al. 2016: Asynchronous Methods for Deep Reinforcement Learning. https://arxiv.org/pdf/1602.01783.pdf
> >
> > **Q3: Any idea why TAGI performs better than backprop on some Atari games but worse on others?**
> >
> > Reply: We have not spent time on identifying why a method performs better than another in some cases because
> >
> > 1) Our resources are spread thin across an array of TAGI developments & applications,
> >
> > 2) We do not think that the algorithm presented here is a finality, but only a starting point.
> >
> > **Q4: Was there a reason for not comparing against Azizzadenesheli et al. (2018) and Osband et al. (2016)? Is there a benefit to the proposed method over these other approaches, aside from mathematical rigor?**
> >
> > Reply: Yes, there are two following reasons
> >
> > 1) A Bayesian deep learning approach quantifies the posterior for the parameters of the network. The paper by Azizzadenesheli is not truly a Bayesian neural network as it only considers Bayesian linear regression on the output layer of the network, similarly, Osband et al is a bootstrapped DQN approach and not a Bayesian one. In the literature, we have not found any Bayesian deep learning approach which estimates the posterior for the network’s parameters and which was scalable enough to be applied to the Atari games’ environment.
> >
> > 2) These approaches are the off-policy reinforcement learning (RL) while our goal is to compare TAGI with the gradient backpropagation on the on-policy RL. This has been explicitly mentioned in Section 4 in the manuscript
> >
> > **Q5: Why does TAGI use a horizon of 128 steps, but the backpropagation baseline uses a horizon of 5 steps? Is this a fair comparison?**
> >
> > Reply: Yes it is a fair comparison because the horizon is considered as the hyperparameters and meaning that it requires to be tuned for a specific method in order to achieve the maximal performance. For Atari example, the Proximal Policy Optimization (PPO) method (Schulman et al. 2017) used the horizon of 128 steps, the Trust Region Policy Optimization (TRPO) (Sculman et al. 2015) employed a horizon of 512 steps, and the Advantage Actor Critic (A2C) (Mnih et al. 2016:) used a horizon of 5 steps. In addition, the horizon of 5 steps is proposed by the authors of the original paper where it was tuned along with other hyperparameters. In the context of TAGI,  we found that the values of 128 steps worked the best across five Atari games.
> >
> > References:
> >
> > Schulman et al. 2017:  Proximal Policy Optimization Algorithms https://arxiv.org/pdf/1707.06347.pdf
> >
> > Schulman et al. 2015: Trust Region Policy Optimization http://proceedings.mlr.press/v37/schulman15.pdf
> >
> > Mnih et al. 2016: Asynchronous Methods for Deep Reinforcement Learning. https://arxiv.org/pdf/1602.01783.pdf
> >
> > **Q6: I’m confused about the exploration rates for backprop on Atari. How are the different epsilon values used in the schedule?**
> >
> > Reply:  According to the original paper (Mnih et al. 2016), the exploration rate is sampled from the distribution taking three values $\epsilon_{1}, \epsilon_{2}, \epsilon_{3}$ with probabilities of 0.4, 0.3, 0.3. These three values are annealed from 1 to 0.1, 0.01, 0.5 respectively over the first four millions frames. Further details can be found in Mnih et al. 2016 (Appendix).
> >
> > References:
> >
> > Mnih et al. 2016: Asynchronous Methods for Deep Reinforcement Learning. https://arxiv.org/pdf/1602.01783.pdf

---

> > > ### Author Response · Authors · 2021-11-16
> > > **Regarding your questions**
> > >
> > > **Q7: What does it mean for the initial learning rate to be LogUniform(1e-4, 1e-2)? Does this mean the learning rate is chosen randomly for each seed?**
> > >
> > > Reply: The initial learning rate is sampled from the distribution LogUniform(1e-4, 1e-2) and annealed to 0 during training. See Mnih et al. 2016 (Appendix) for further details.
> > >
> > > References:
> > >
> > > Mnih et al. 2016: Asynchronous Methods for Deep Reinforcement Learning. https://arxiv.org/pdf/1602.01783.pdf
> > >
> > > **Q8: The exploration fraction (0.1)---which I assume is the same thing as epsilon---for backprop on CartPole and Lunar Lander seems rather high. Are you decaying that to 0.02? If not, how was this value chosen?**
> > >
> > > Reply: As mentioned explicitly in the section 4.1, we employed the same hyperparameters as  the well-tuned implementation from openai baselines (Dhariwal et al. 2017).
> > >
> > > References:
> > >
> > > Dhariwal et al. 2017: Openai baselines https://github.com/openai/baselines
> > >
> > > **Q9: It’s interesting that the variance for TAGI seems to be much lower than for backprop on CartPole and Lunar Lander, but much higher on Atari. Does this hold up with more seeds? What might explain this behavior?**
> > >
> > > Reply: Yes it does. The additional results were added to the manuscript. We think that the environment is different from each other. Lower variance in the cart pole and lunar lander does not mean that we will observe the same behavior in the Atari environment.
> > >
> > > **Q10: Is there a potential connection between entropy regularized RL and this notion of using a decay function for the observation noise variance to more strongly incentivize the prior during early learning?**
> > >
> > > Reply. We have not investigated entropy regularized RL.

---

> > > > ### Comment · Reviewer_Aorz · 2021-11-30
> > > > **Reviewer Response**
> > > >
> > > > Thank you for the response, and for taking the time to answer these questions.
> > > >
> > > > I understand and appreciate the value of applying Bayesian techniques to reinforcement learning, and I think that is a worthwhile line of research to pursue. The broader point that I was trying to make with my review was that the paper can be interpreted in a number of different ways. Focusing on any one of these aspects might make for an acceptable paper if done well. However, the paper does _not_ focus on any one of them sufficiently to tell a clear story, and has a number of significant weaknesses in its present form.
> > > >
> > > > With my review, I tried to give a sense of what I thought the main weaknesses are, as well as the questions that I was left with that I felt the paper itself did not sufficiently answer. While I wouldn't necessarily expect the paper to answer all of these questions, I would expect it to answer at least some of them, or to provoke other, different questions and answer those instead.
> > > >
> > > > Again, I think this is an interesting and valuable research direction, but I feel the paper is not ready for publication in its current state.

---

### Official Review · Reviewer_H4dH · 2021-11-02

**Correctness:** 4
**Technical Novelty And Significance:** 2
**Empirical Novelty And Significance:** 2
**Recommendation:** 3
**Confidence:** 2

**Main Review:**

Strength:

This paper is well-written and easy to follow. The idea of applying Bayesian inference to obtain uncertainty estimation and sampling is interesting.

Weakness:

Major concern:

Since the key idea of TAGI-DQN, the TAGI algorithm, is from an existing paper, I'm expecting more complete empirical evaluations and more systematic comparisons with the baseline model (gradient-based algorithm), and the current experiment part may not be enough.
1. The comparisons should be implemented on more Atari games with more epochs since Atari games are very different for each task and may have different learning patterns in the early stage of training. (as shown in paper [1])
2. The performances of DQN here is much lower than the ones shown in other paper (e.g., [1,2]), which makes the conclusion less convincing.

It would be more persuasive if adding TAGI-DQN on top of the codebase that has better performances, and run through more Atari games to conclude that TAGI-DQN can achieve similar performances with its gradient-based counterpart.

Besides, the assumption homoscedastic variance of TAGI-DQN is not very realistic since the variance of $V$ at step $0$ should be very different from that at step $T$ (last step), which requires more strong empirical results to support.

3. I think the focus of this paper is a little bit unclear to me. In the related work part, it seems the main contribution of TAGI-DQN is to propose an analytically tractable inference approach; whereas in the Discussion part, the main focus is to propose a gradient-free method compared with its backpropagation counterpart. If the first is the case, then I would like to see more comparisons between the existing BDL methods; if the latter is the case, I think the related work part should be modified to reflect more from this aspect.

Minor concern:

1. The definition of A in eq(4) is not the same as the size of action space.
2. It is not very clear to me what does "Atari experiment with on-policy RL" mean, since Q-learning is an off-policy algorithm, what does on-policy indicate here?
3. I do not see how eq(10) takes the maximization of the expected value as described one line above.

1: Hessel, Matteo, et al. "Rainbow: Combining improvements in deep reinforcement learning." Thirty-second AAAI conference on artificial intelligence. 2018.

2:Dabney, Will, et al. "Distributional reinforcement learning with quantile regression." Thirty-Second AAAI Conference on Artificial Intelligence. 2018.


**Summary Of The Paper:**

This paper combines the TAGI algorithm with Q-learning algorithm and demonstrates that TAGI-DQN can achieve on-par performance with its backpropagation-based counterpart with fewer hyperparameters.


**Summary Of The Review:**

In summary, the idea of applying Bayesian inference as an alternative to backpropagation-based algorithm is interesting. However, it requires more complete experiments to demonstrate this, along with the other assumptions that simplify the computing complexity.

---

> ### Author Response · Authors · 2021-11-16
> **Rebuttal and Clarifications**
>
> Although adding more games and training for longer could be interesting, we do not have the resources required at the moment. Because TAGI is fundamentally different from any other backprop-trained approach is it incompatible with existing library, e.g. PyTorch and Tensorflow, and this require extensive amount of developpement. Moreover, note that we do not compare with the latest architecture because it is not the point of the paper; Here, The main objective of our paper proposes a new framework that couples the temporal difference Q-learning with the Tractable Approximate Gaussian Inference (TAGI).  More specifically, TAGI provides the estimation of the model parameter uncertainty using closed-form Bayesian inference, and the Thompson sampling then factors the model parameter uncertainty into the selection of an action. The potential of the proposed framework is then illustrated on complex environments such as Atari. In our opinion, this itself is a valuable contribution as TAGI is the first closed-form analytically tractable inference method that, as shown with this paper, scales for deep learning models.

---

### Official Review · Reviewer_6wux · 2021-11-02

**Correctness:** 3
**Technical Novelty And Significance:** 3
**Empirical Novelty And Significance:** 2
**Recommendation:** 3
**Confidence:** 3

**Main Review:**

The paper presents an interesting new solution to the problem of performing Bayesian deep RL, which is certainly a challenging problem for which a lot of use can be seen. As they note, the associated uncertainty is certainly very useful for driving appropriate strategies for the agent to balance exploration and exploitation. However I don't think the paper is quite ready to be published.

Firstly, I'm not sure this citation is correct:
J-A. Goulet, L.H. Nguyen, and S. Amiri. Tractable approximate Gaussian inference for Bayesian
neural networks. Journal of Machine Learning Research, 22:1–23, 2021.
I can't find the paper on the JMLR website (https://jmlr.org/papers/v22/), and as far as at least Google Scholar is showing it only seems to exist on arxiv. Of course, if I just haven't looked sufficiently hard enough (or in the wrong place) please correct me. This may seem like an odd complaint for the main review, but since the method presented in the paper is a direct extension of this TAGI method to deep RL it's important to be clear where the starting point is. This makes it a little harder to validate the claims made about the underlying TAGI method, since as far as I can see it does not appear to have been peer-reviewed yet. Having looked at it myself briefly, I will assume for the purposes of this review though that the claims hold.

In the experimental section there are no comparisons made to any other Bayesian deep RL methods despite them being discussed in the paper. This seems to be justified by saying that the presented method is sufficiently special and distinct because it is not a gradient based method. I'm not really sure why that's particularly important, but even so I think it very useful/necessary to have comparisons with those methods anyway so as to see the relative performance.

The comparison is always made to a deterministic neural network but importantly following a $\epsilon$-greedy exploration strategy, as opposed to the Thompson sampling done by this method. Other exploration strategies are available (upper confidence bound for example) and I think should all be considered (or at least explained why not considered) in the paper and experiments.

A large deal is made of the fact that no gradients are required - why this is such a major contribution should probably be discussed furthering the paper as I don't think that it's made particularly clear at the moment.

I really appreciate the use of the colours in the figures - I'm not sure though whether magenta and grey are the best choices as they don't particularly stand out.



**Summary Of The Paper:**

The authors present a method for combining deep Q-learning with tractable approximate Gaussian inference to allow for deep Bayesian reinforcement learning without the need for gradient based updates. They demonstrate this method can attain performance comparable to standard backprop trained agents in a number of environments.

**Summary Of The Review:**

The paper presents a potentially promising method for Bayesian deep RL that can scale to large states spaces like Atari problems. I do not however believe that the paper is ready to be published as is since the experimental validation and comparison needs expanding, and further discussion on why exactly this method is so useful compared to the current literature would be useful.

---

> ### Author Response · Authors · 2021-11-16
> **Rebuttal and Clarifications**
>
> Regarding the reference to Goulet, Nguyen, and Amiri (2021), the citation is incorrect as the paper is accepted but not yet published in JMLR.
>
> We did not compare with Bayesian deep RL methods simply because we could not find any Bayesian deep learning approach which estimated the posterior for the network’s parameters and which was scalable enough to be applied to the Atari games’ environment.
>
> Although it is clear to us, we now see after reading through the reviewers’ comments that you do not perceive the intrinsic value of performing closed-form Bayesian inference instead of relying on gradient-based optimization. Given that distance between our and your perception, we are not trying to convince you to change your mind about the manuscript. That being said, it is already widely acknowledged that characterizing the Bayesian posterior for the parameter is the best approach [Goodfellow, Bengio, and Courville (2016), Murphy (2012)]. Up to now the Bayesian approach is almost never employed simply because the estimating the posterior in large models was computationally intractable; As we show in this paper, with TAGI, it is not anymore the case. The main objective of our paper proposes a new framework that couples the temporal difference Q-learning with the Tractable Approximate Gaussian Inference (TAGI).  More specifically, TAGI provides the estimation of the model parameter uncertainty using closed-form Bayesian inference, and the Thompson sampling then factors the model parameter uncertainty into the selection of an action. The potential of the proposed framework is then illustrated on complex environments such as Atari. In our opinion, this itself is a valuable contribution as TAGI is the first closed-form analytically tractable inference method that, as shown with this paper, scales for deep learning models.
>
>
> Reference:
>
> J-A. Goulet, L.H. Nguyen, and S. Amiri. Tractable approximate Gaussian inference for Bayesian neural networks. Journal of Machine Learning Research, 22:1–23, 2021 (accepted for publication, http://profs.polymtl.ca/jagoulet/Site/Papers/2021_Goulet_Nguyen_Amiri_TAGI_JMLR.pdf).
>
> Goodfellow, Bengio, and Courville (2016), Deep Learning, MIT Press. §5.6.1
>
> Murphy (2012), Machine Learning – A probabilistic perspective, MIT Press. §5.2.1

---

> > ### Comment · Reviewer_6wux · 2021-11-17
> > **Reviewer Response**
> >
> > Maybe I didn't make this clear in the main review, but I am not that adversarial - I think fundamentally the algorithm is good, and in general is a reasonable research question that would merit publication. However, I think that as a research paper it simply needs to be better in motivating the setting and properly compare with the current literature (which at the moment is completely brushed under the carpet). With those changes I would have little issue recommending acceptance. With that being said, with respect to your comments:
> >
> > Whether or not the current Bayesian deep RL methods scale to Atari game environments is beside the point - it is still appropriate to show (in potentially simpler environments) how TAGI compares to these other directly related methods so that readers of your paper have a better idea how this method fits into the literature. Clearly though you do not see the intrinsic benefit of comparing to other methods in the literature so I won't try any further to convince you of such.
> >
> > "Although it is clear to us, we now see after reading through the reviewers’ comments that you do not perceive the intrinsic value of performing closed-form Bayesian inference instead of relying on gradient-based optimization"
> > This is a total straw man of my position, and I cannot see how it follows from my review or is an appropriate response. In fact my issue is not *closed-form Bayesian inference* vs *gradient-based optimization* but rather *closed-form approximate Bayesian inference* vs *gradient-based approximate Bayesian inference*. Regardless - even if this was the case, it is **your** job as authors to explain this and not rely on the intuition of readers.
> >
> > "it is already widely acknowledged that characterizing the Bayesian posterior for the parameter is the best approach" What exactly are you saying here? If you are arguing that a Bayesian approach in general is appropriate then I am all on board - if you are saying that what is needed is some parametric from of the posterior to analyse (as opposed to say MC samples) then I am slightly more skeptical but also inclined to agree. Regardless, two references to some of the biggest and most general textbooks in machine learning are not appropriate sources upon which to make specific claims about characterising a posterior distribution. In any case though - this is a separate issue to the question of closed form vs gradient updating. Estimating the posterior using gradient methods in large Bayesian neural nets is not remotely intractable using the variety of variational approximate methods which will still produce a "characterized" Gaussian posterior from which you can then sample, and the application of Thompson sampling to factor "the model parameter uncertainty into the selection of an action" is not new, that is the point of Thompson sampling. Since it is not the case that TAGI is an *exact* solution, it still makes heavy assumptions the approximate form of the posterior distribution - why then exactly should I care about the fact that TAGI is done in a closed form manner? In fact, it only makes it harder to implement, as you point out in your response to Reviewer H4dH! Perhaps there is a computational saving in certain settings? Well then that should be explained clearly but again is not actually an intrinsic benefit of closed form solutions.

---

### Official Review · Reviewer_vNfB · 2021-11-09

**Correctness:** 2
**Technical Novelty And Significance:** 2
**Empirical Novelty And Significance:** 3
**Recommendation:** 3
**Confidence:** 2

**Main Review:**

Strengths:
1. This paper applies a Bayesian inference methods to do reinforcement learning and shows effectiveness on Atari level benchmarks.
2. It is potentially useful in the settings when one need to do deep RL without using gradient based optimization.

Weaknesses:
In general, the algorithm seems largely rely on an existing algorithm, TAGI, and the main contribution is extending this to the deep Q learning settings. In this case, I would expect a much better written paper in explaining the algorithm/contribution and stronger empirical results.
1. The property that it use Bayesian inference (gradient free) to update the NN's parameters seems to be an important contribution. First, it need to be better explained why no gradient itself is a pro. I would take it as just a property of an algorithm, without more advantage of disadvantage. Second it is very hard for me to understand the full update rule of each layer's parameter after reading this paper and its appendix. It might due to my limited knowledge in Bayesian inference, but I think this paper could be more self-contained by adding the full derivation of each layer's neural networks.
2. Most of the theoretical assumptions are not well explained and justified. For examples, what is the definition of A3 in math; why A5 is reasonable. In particular, one assumption about that different values has the same scale of variance seems very strong, especially in multi-step settings. Often the decisions not only effect the expectation, but also the variance (and distribution) of the return. Many distributional RL, safe RL literature make use of this and shows benefit.
3. It surprised me that this paper does not include any Bayesian deep RL baselines. Though some of them might not be gradient free like this paper. They shares many similarities such as the Gaussian assumptions.

**Summary Of The Paper:**

This paper proposed a new Bayesian Deep RL algorithm that learns deep Q networks (DQN) without using gradient descent. The updates of DQN's parameters follows from the tractable approximate Gaussian inference (TAGI) algorithm. Then it studies the empirical performance of proposed algorithm with DQN learned by gradient based optimization in several Atari benchmarks.

**Summary Of The Review:**

In summary, this paper studied how to learn DQN in a fully Bayesian way and is interesting. However too many parts of the paper related to its core contribution is unclear at this time. Unless it's significantly updated then this paper is not ready to published.

---

> ### Author Response · Authors · 2021-11-16
> **Rebuttal and Clarifications**
>
> It is clear after reading through the reviewers’ comments that you do not perceive the intrinsic value of performing closed-form Bayesian inference instead of relying on gradient-based optimization. The main objective of our paper proposes a new framework that couples the temporal difference Q-learning with the Tractable Approximate Gaussian Inference (TAGI).  More specifically, TAGI provides the estimation of the model parameter uncertainty using closed-form Bayesian inference, and the Thompson sampling then factors the model parameter uncertainty into the selection of an action. The potential of the proposed framework is then illustrated on complex environments such as Atari; In our opinion, this itself is a valuable contribution as TAGI is the first closed-form analytically tractable inference method that, as shown with this paper, scales for deep learning models.
>
> The justification for A3 and A5 are detailed in Goulet, Nguyen, and Amiri (2021).
>
> Regarding the “assumption about that different values has the same scale of variance”, we are currently working on relaxing this assumption. If the assumption turns out to be key, the results should only end-up being better than they currently are.
>
> We did not include any “Bayesian deep RL baselines” simply because we could not find any Bayesian deep learning approach which estimated the posterior for the network’s parameters and which was scalable enough to be applied to the Atari games’ environment.
>
> Reference:
>
> J-A. Goulet, L.H. Nguyen, and S. Amiri. Tractable approximate Gaussian inference for Bayesian neural networks. Journal of Machine Learning Research, 22:1–23, 2021 (accepted for publication, http://profs.polymtl.ca/jagoulet/Site/Papers/2021_Goulet_Nguyen_Amiri_TAGI_JMLR.pdf).

---

### Decision · Program_Chairs · 2022-01-20

**Decision:**

Reject

**Comment:**

The authors combine TAGI with Q-learning to create an approximate Bayesian Q-learning algorithm. They evaluate their approach and show it has comparable performance to DQN.

All of the reviewers were positive about the potential of the paper. Unfortunately, the paper suffers from lack of clarity, of motivation, and comparison to relevant approaches. All of the reviewers brought up nearly the same concerns and I agree with these concerns. The authors have not address them and the reviewers do not think this paper is ready for publication at this time. I agree and recommend rejection.

Evaluating TAGI-DQN is a valuable contribution, but alone it is not sufficient. The reviewers have made many suggestions on how to improve the paper, and I hope the authors follow up on these suggestions.